# Electronic Personal Health Records for Mobile Populations: A Rapid Systematic Literature Review

**DOI:** 10.3390/ijerph22040488

**Published:** 2025-03-25

**Authors:** Paulien Tensen, Francisca Gaifém, Simeon Kintu Paul, Frederick Murunga Wekesah, Princess Ruhama Acheampong, Maria Bach Nikolajsen, Ulrik Bak Kirk, Ellis Owusu-Dabo, Per Kallestrup, Charles Agyemang, Steven van de Vijver

**Affiliations:** 1Department of Public and Occupational Health, Amsterdam Public Health, Amsterdam UMC, University of Amsterdam, 1105 AZ Amsterdam, The Netherlands; p.tensen@amsterdamumc.nl (P.T.); c.o.agyemang@amsterdamumc.nl (C.A.); 2Amsterdam Health & Technology Institute, 1105 BP Amsterdam, The Netherlands; svijver@gmail.com; 3Research Unit for Global Health, Aarhus University, 8000 Aarhus, Denmark; 4African Population and Health Research Center, Nairobi P.O. Box 10787-00100, Kenya; simeonkintu@gmail.com (S.K.P.); fwekesah@aphrc.org (F.M.W.); 5Julius Global Health, Julius Center for Health Sciences and Primary Care, University Medical Center Utrecht, Utrecht University, 3584 CS Utrecht, The Netherlands; 6Department of Health Promotion and Disability Studies, School of Public Health, College of Health Sciences, Kwame Nkrumah University of Science and Technology, Kumasi 00233, Ghana; princess.acheampong@knust.edu.gh; 7Social Inequality, Research Unit for General Practice, 8000 Aarhus, Denmark; mbn@ph.au.dk; 8Digital Health, Research Unit for General Practice, 8000 Aarhus, Denmark; ubk@ph.au.dk; 9Department of Public Health, Aarhus University, 8000 Aarhus, Denmark; per.kallestrup@ph.au.dk; 10Department of Global Health, School of Public Health, College of Health Sciences, Kwame Nkrumah University of Science and Technology, Kumasi 00233, Ghana; eowusu-dabo.chs@knust.edu.gh; 11Division of Endocrinology, Diabetes, and Metabolism, Department of Medicine, Johns Hopkins University School of Medicine, Baltimore, MD 21205, USA; 12Department of General Practice, OLVG Hospital, 1091 HA Amsterdam, The Netherlands

**Keywords:** electronic personal health record, migration, mobile populations, medical data exchange, digital health, health equity

## Abstract

**Background:** Mobile populations, including refugees, asylum seekers, and undocumented migrants, face challenges related to access, continuity, and quality of healthcare, among others, due to the lack of available health records. This study aimed to examine the current landscape of Electronic Personal Health Records (EPHRs) developed for and used by mobile populations. **Methods:** A rapid systematic review was conducted between September 2024 and January 2025, identifying relevant publications through searches in Embase, PubMed, Scopus, and grey literature. **Results:** The literature search yielded 2303 articles, with 74 remaining after title and abstract screening. After full-text screening, 10 scientific articles and 9 grey literature records were included in a qualitative data synthesis. Six distinct EPHRs were identified, differing in how they centralize health records, in additional functionalities, and the level of patient autonomy granted. **Discussion and Conclusions:** Limited evidence exists on EPHRs impact on health outcomes or continuity of care, and user adoption remains a critical challenge. Key elements in the development and implementation of EPHRs include ensuring a high level of data security and co-designing easy-to-use EPHRs. The review indicates a need for future research on user experiences of EPHRs and their impact on the health outcomes of mobile populations.

## 1. Introduction

### Rationale

Migration is a common phenomenon in the history of mankind. Migration is broadly defined as “the movement of persons away from their place of usual residence”, irrespective of reasons for the movement and means or routes used to migrate [1].

While the majority of migration is regular, safe, orderly, and often directly related to work [2], the reality of migration is complex. Current estimates indicate that there are more than 250 million international migrants in the world accounting for approximately 3.6 percent of the global population [3]. Moreover, in the last decade, there has been a significant increase in forced displacement due to conflict, violence, persecution, political or economic instability, climate change, and other disasters. By the end of 2023, the global population of displaced individuals exceeded 117 million, including around 68 million internally displaced persons, 38 million refugees, and 7 million asylum seekers [3].

In this article, we use the term mobile populations to refer to individuals or groups of persons who, whether voluntarily or forcibly, change their location of residence. This definition includes movements that occur locally, nationally, and/or across borders, for a short and/or long duration [1]. While this broad definition intends to encompass the complexity of current migration trends and recognizes challenges that cut across different groups of mobile populations, our review deals specifically with displaced individuals, asylum seekers, refugees, stateless persons, and undocumented migrants (UDMs)—mobile groups that face more significant disadvantages and are most in need.

Migration is considered a determinant of health and there is a strong link between migration and access to quality healthcare [4]. Even though healthcare is widely recognized as a fundamental human right [5], certain groups of mobile populations, such as refugees, frequently encounter significant barriers to accessing healthcare services and ensuring continuity of care [6]. Barriers include sociocultural and language differences, lack of information on where to obtain care, economic barriers, and restrictive regulations to access healthcare without valid legal status [7].

Among these barriers, limited access to health records presents a serious challenge to ensuring continuity and quality of care for mobile populations [8]. Health records typically contain information about a patient’s medical history, such as diagnoses, medical charts, and treatment plans, which are important for informed clinical decision-making [9]. A key reason for the lack of access to health records is that mobile populations change geographical locations, and therefore are treated by multiple healthcare providers (HCPs) across various sites [10]. Health records from both previous places of residence and those generated along migration routes are seldom shared, often due to incompatible or non-existing health record systems. Health records are also rarely shared with the patients themselves [11]. A lack of access to health records also presents significant challenges for HCPs, resulting in duplicative diagnostic procedures and healthcare delays, which in turn affect the health outcomes of mobile populations [7].

There are over five billion mobile phone subscribers worldwide (77% of the world population) [12]. With the figures significantly higher in high-income regions, the numbers are also steadily increasing in low- and middle-income countries (LMICs), where there is roughly 68% penetration of mobile telephone coverage [12]. Additionally, mobile phone usage among mobile populations has grown significantly in recent years [13]. Mobile populations, particularly those from LMICs, often rely heavily on mobile phones as they are a cost-effective means of staying connected with family and friends and accessing information [14]. According to a 2019 survey by the International Organization for Migration (IOM), around 80% of migrants use mobile phones to communicate with family members, seek job opportunities, and stay informed about their legal status. As of 2023, there were an estimated 300+ million mobile money accounts globally, with a significant portion of those users being migrants who send remittances to their home countries [12].

There is also a growing use of smartphones among mobile populations, which provides opportunities to improve access to healthcare. Several digital health initiatives exist aimed at migrants and refugees with the intention to provide health information and tools that assist in facilitating access to care and communication with HCPs [15,16]. Among these initiatives, electronic health records (EHRs) developed for displaced populations have demonstrated the potential to improve patient health outcomes and increase the efficiency of collecting and accessing health information from patients [17]. Electronic Personal Health Records (EPHRs) in particular have received increased attention due to their potential to empower patients, improve care coordination, reduce healthcare costs, and promote better health outcomes [18]. EPHRs are defined by the Markle Foundation (2004) as “electronic systems that allow individuals to access, manage, and share their health information in a confidential and secure environment that allows users to coordinate their lifelong health data and share relevant portions with those who need it” [19]. Typically, EPHRs include information regarding an individual’s medical history, diagnostic results, treatment plans, allergies, and immunizations, with functionalities to update records as new information becomes available. EPHRs—unlike traditional health records owned, accessed, and maintained by HCPs—may empower users with greater control over their health records, enabling them to take a more active role in their healthcare management [9]. Essen et al. (2018) describe that knowledge of—and access to—one’s own medical history is important to access and benefit from good quality healthcare [20].

By bridging gaps in health data accessibility, EPHRs have great potential to improve continuity of care for mobile populations [6], especially informational continuity. This could be specifically valuable for mobile populations who often interact with multiple HCPs across different locations [10], especially for specific clinical areas such as reproductive health issues, infectious diseases, and chronic diseases, which are highly relevant for mobile populations [18,21]. There is potential for EPHRs to promote continuity of care by serving as a central repository for health data that can be accessed and shared regardless of geographic location.

In addition, EPHRs support patient-centred care by providing a centralized and comprehensive approach to health data management. A scoping review found that patients who managed their health records felt more knowledgeable and in control of their own healthcare process, and therefore felt more prepared for their clinical visits [22]. Access to one’s medical records may promote greater engagement of patients in their treatment decisions and improve medical information sharing with HCPs, leading to more informed decision-making, less redundant testing, and fewer medical errors including misdiagnosis [6,22,23].

Despite these potential advantages, the development and implementation of EPHRs remain limited due to barriers such as the lack of understanding of the importance of access to personal medical data, limited digital literacy, concerns about data security, and privacy, and the financial and technical challenges of implementation [9]. However, as the field evolves, innovations in technology such as mobile health apps and cloud-based systems offer promising opportunities to overcome these challenges and make EPHRs a cornerstone of healthcare for mobile populations [24].

Existing systematic reviews have investigated the potential role of EPHRs in improving healthcare and health outcomes [6,23,25,26]. However, no review has specifically focussed on EPHRs for mobile populations. Furthermore, the most recent systematic review on health records for a subgroup of mobile populations, migrants, and refugees, was published in 2019. Significant digital innovations have emerged since.

This rapid systematic literature review examines the current landscape of EPHRs designed for mobile populations. This review aimed to identify and describe EPHRs that have been developed for and utilized by these groups. Additionally, it will explore user experiences with EPHRs, investigate facilitators and barriers to their implementation, and highlight the strengths and weaknesses of the existing EPHRs available for mobile populations.

## 2. Materials and Methods

A rapid systematic literature review protocol was developed according to the Preferred Reporting Items for Systematic Reviews and Meta-Analyses (PRISMA) guidelines (Appendix A). The protocol was registered on the International Prospective Register of Systematic Reviews (PROSPERO) with registration number CRD42024604242.

### 2.1. Search Strategy

Relevant scientific publications were identified using a search query developed in English and cross-checked by a librarian. The query was run on three scientific databases (EMBASE, PUBMED, and SCOPUS), in October 2024, for articles published between 2014 and 2024. The year 2014 was chosen as the cut-off based on three factors: previous published reviews on EPHRs, the more substantial investment in digital health innovations witnessed from this year onwards, and the more significant influx of migrants and refugees to Europe, which heightened attention to migration. The detailed search queries used for each database are provided in Appendix B. Additionally, one article that was identified as relevant prior to the search was included, despite not being retrieved through the search strategy.

### 2.2. Study Selection

The publications resulting from our primary search were uploaded to Rayyan.ai (Doha: QCRI), a free online Systematic Review Management Platform. Duplicates were subsequently removed manually. Four reviewers were involved in screening, and all remaining articles were independently screened by two reviewers on title and abstract. The same blinded reviewers thereafter screened the selected full texts for potential eligibility. Any conflicts were discussed and resolved among the reviewers.

The inclusion and exclusion criteria for articles in this review were defined a priori. Articles were included if they adhered to the following criteria: (1) were published in English; (2) were published between January 2014 and October 2024; (3) described an EPHR for mobile populations; (4) explored EPHRs as an intervention or outcome; (5) described EPHRs which were still existent; (6) described EPHRs which were not restricted to a specific medical condition; and (7) described EPHRs which included a digital component that allowed the patient to manage their health records. This set of criteria led to the exclusion of publications if the described tool was only used as a data source to collect quantitative outcomes for a study, if the tool was an existing facility-based system and if the tool was developed mainly for HCPs in a specific health clinic. Moreover, studies were excluded when no full text was available. Inclusion criteria were applied sequentially. To include any tool having a personal component to health data management, but perhaps not named as an EPHR in the title/abstract, we retained articles referencing EHR in their title/abstract and narrowed our inclusion criteria during full-text screening once enough detail was provided to match our focus on EPHRs. Literature reviews were excluded, but no further restriction on the type of article was applied. Articles included in the literature reviews that met our inclusion criteria were screened individually, to make sure that relevant articles were not missed. No restrictions were set on the study design, and/or study setting.

### 2.3. Data Extraction

Data were extracted by the lead authors, with support from two co-authors, using a standardized extraction form that was piloted with one article and adjusted iteratively during the data extraction process. All data were extracted by two independent reviewers. The data extraction form was filled out in MS Excel.

Information extracted from the included articles encompassed various study characteristics, such as design, setting, population, aim, methodology, and duration. It also included details about the digital health application (tool), the information or data captured by the tool, the number of users, stakeholders involved in its development, the stage of development, elements related to data entry and processing, data protection measures, the study population served by the application, and user engagement/experiences.

### 2.4. Grey Literature

Though not always included in rapid methods for systematic reviews, in the present rapid systematic review we included grey literature to address information gaps in the data extraction form and to identify initiatives that were not covered in peer-reviewed literature. Grey literature searches were concluded in January 2025.

Based on experience in the field of EPHRs for mobile populations, the researchers were familiar with several potential EPHR initiatives. Moreover, two key experts in the field of EPHRs, from Canada and the United Kingdom, were asked via email to share initiatives they were familiar with. The Google Search Engine was used by combining different terms from the search queries in the scientific literature. Inclusion criteria for the grey literature were official websites of EPHR tools, government and university webpages, reports, technical notes, and information derived from the Play Store on the selected tools. Moreover, an inclusion criterion was the availability of sufficient information about a tool to allow for analysis. For tools where information still was very limited, the entities responsible for the tools were contacted via email for the specific information missing.

### 2.5. Synthesis of Results

Due to the heterogeneity of the included publications, a qualitative narrative synthesis was conducted but it was not possible to conduct a statistical meta-analysis.

### 2.6. Ethical Review Statement

No ethical review was necessary as data were only collected from online databases and grey literature that was publicly available. This research also did not involve any human subjects.

### 2.7. Patient and Public Involvement Statement

No patients or the public were involved in the design, conduct, reporting or dissemination plans.

## 3. Results

### 3.1. Literature

The scientific literature search yielded a total of 2303 records, which resulted in 74 relevant publications after the initial screening of the title and abstract and the elimination of duplicates. Publications were screened as illustrated in Figure 1. After the subsequent screening based on the full text, 10 peer-reviewed articles and 9 grey literature records were included in the narrative synthesis.

#### 3.1.1. Results from Eligible Peer-Reviewed Publications

Details of the 10 peer-reviewed articles included in this rapid systematic review are provided in Table 1. All 10 articles were published between 2018 and 2024. Most of the included articles did not follow a specific study design, except for one cross-sectional study and two qualitative studies. One publication was a report, while the remaining ones had a descriptive nature, including opinion papers, comments and viewpoints. As depicted in Table 1, several of the scientific articles focused on the same tool. Three articles focused on the electronic Maternal and Child Health Handbook (e-MCH) application, and Sijilli (meaning “my record” in Arabic), two articles focussed on the HERA App (the Health Recording application), and one article focussed on HealthEmove. Since most eligible articles provided only narrative evidence, a quality assessment of the included studies based on critical appraisal guidelines was not deemed relevant.

#### 3.1.2. Results from Grey Literature

The grey literature search provided additional sources of information to the systematic review. These included official websites of the identified tools/applications (HealthEmove, HERA App, RedSafe, My Personal Health Bank), LinkedIn pages of the identified tools (My Personal Health Bank), webpages of universities (My Personal Health Bank), technical briefs (e-MCH Handbook), news releases (RedSafe), and information from the Play Store (RedSafe). All grey literature sources are documented in Table 2.

### 3.2. Initiatives

In total, six tools were identified and included in this review. Table 3 describes the main characteristics of these tools. Several tools which initially appeared to meet the inclusion criteria, were nevertheless excluded. CARE was excluded due to a lack of information on programme continuity after its conclusion [21]. PANDA, Re-health2, and HIKMA health were provider-focused systems with no evidence of, or information on, patient access to health information via an application [21,45,46]. The e-NCD application lacked sufficient information for inclusion [35], and no English information was found on the continuity of Prevenzione 4.0 after 2014–2020 [47]. While Sana.NCD included a patient-controlled health record, details on this were unavailable, and publications focused only on a clinician tool [48]. CImA focused solely on vaccination data [49], and Ucraid comprised two applications targeting allergies, asthma, and chronic urticaria exclusively [50]. Finally, Univie was excluded after consulting a co-founder who clarified that the project is no longer running [51].

#### 3.2.1. Main Characteristics of the Tools Identified

##### Mobile Populations

Three of the tools identified were specifically developed for and used by refugees (e-MCH Handbook, Sijilli, and HERA App). The remaining three tools had broader target populations, defined as ‘people on the move’ (HealthEmove and My Personal Health Bank), ‘developing countries’ (My Personal Health Bank) and ‘people affected by conflict, migration and other humanitarian crisis’ (RedSafe). While most tools focus on mobile populations in general, the e-MCH Handbook and the HERA App have a specific focus on pregnant women and mothers [28,30]. Three of the tools were also specifically developed to serve refugees: Palestinian refugees in the case of the e-MCH Handbook and Syrian refugees in the case of the HERA App and Sijilli [31,33,35].

##### Countries Covered, Stage of Development, and Number of Users

In total, the tools were present in 24 countries. Of these, four countries are located in the broader ‘Global North’ (The Netherlands, Switzerland, USA, and Turkey). The remaining countries belong to the WHO Eastern Mediterranean Region (Gaza, Jordan, Lebanon, Syria, Westbank), the WHO Region of the Americas (Costa Rica, El Salvador, Guatemala, Honduras, Mexico, Panama), and the WHO African Region (Botswana, Eswatini, Lesotho, Malawi, Mozambique, South Africa, Tanzania, Zambia and Zimbabwe).

The tools identified in this study are at various stages of development and implementation. The e-MCH Handbook, the HERA App, RedSafe, and Sijilli are at the implementation phase. Two of these tools are implemented in multiple countries (e-MCH Handbook, and RedSafe), while the other two tools are implemented in a single country (Sijilli, and the HERA App). The e-MCH Handbook has been available since 2017, with 254,586 Palestine refugee mothers or pregnant women registered and 22,000 individuals actively using the tool in 2023 [35]. The HERA App, specifically designed for the Syrian refugee population in Turkey, was launched in 2018 [29]. According to its official website, the HERA App is being used by more than 3000 refugee families [37]. The HERA App has been field-tested by the Medical Rescue Association of Turkey (MEDAK), a grassroots organization [32]. There is an ambition to implement the HERA App for the whole Syrian refugee population in Turkey [32]. By February 2022, RedSafe had been downloaded by 32,000 people ‘affected by conflict, migration or by a humanitarian crisis’, across the 15 countries in which the application is available [38,39]. Sijilli was launched in 2018 and by 2020 more than 10,000 Syrian refugees in Lebanon were using this tool [31].

HealthEmove and My Personal Health Bank are in the pilot phase and were each piloted in a single country, the Netherlands and Tanzania, respectively [36,41]. There are 4969 users from six different health centres and hospitals in Tanzania on My Personal Health Bank [41]. No information was found on the number of users of HealthEmove. HealthEmove intends to expand to other European countries, while My Personal Health Bank aims to expand to LMICs, starting in Tanzania and Rwanda [36,41].

##### Languages

The tools also support multiple languages to meet the needs of different populations. All tools except e-MCH Handbook are available in English. The e-MCH Handbook, the HERA App, Sijilli, and HealthEmove are available in Arabic [27,29,31,36]. Additionally, the HERA App supports Turkish, Pashto, and Dari [37]. My Personal Health Bank is also available in Kiswahili [41], RedSafe, and HealthEmove also offer Spanish and Portuguese [38] and HealthEmove is available in 22 languages in total [36].

##### Tool Description

While all six tools can centralize health records, the aim of the tools—and the extent to which they focus on health record-keeping—varied significantly.

The HERA App and RedSafe are both considered humanitarian platforms and are available as smartphone apps. In addition, RedSafe can also be used as a web app. Both tools allow access to health information and information about the nearest humanitarian or health services [32,38]. Moreover, the HERA App includes information about first aid, information about the Turkish healthcare system [33], and provides appointment reminders for pregnant women and child vaccinations [32]. These appointment reminders include the dates of vaccinations for children according to the Turkish Vaccination Calendar, and antenatal checkup reminders [32]. Both RedSafe and the HERA App include a ‘digital vault‘ to store medical documents, such as photos of medical documents, and both include a Geographic Information System (GIS) tool to locate humanitarian aid services [37,38]. RedSafe extends its functionality to assist users in locating family members, and allow users to send messages to people in the RedSafe directory, while the HERA App allows users to contact emergency services, includes a WhatsApp chat function, and can be used to call emergency services [37,38,39].

The e-MCH Handbook is an electronic version of the Maternal and Child Health (MCH) Handbook, used by Palestine refugee mothers since 2008 [52]. The e-MCH Handbook is linked to the EHR system of the United Nations Relief and Works Agency for Palestine Refugees in the Near East (UNRWA). The app is used to send appointment reminders to users, it can be used to communicate with UNRWA health centres, and it provides access to educational information and personal health records of pregnant women, mothers, and their children [27,30,35].

Sijilli, HealthEmove, and My Personal Health Bank are health record systems with the primary goal of capturing essential health information of individuals electronically. These tools allow individuals to transport their health records across settings, using a web app (Sijilli, HealthEmove, and My Personal Health Bank) or a physical USB stick (Sijilli) [16,31,36,41]. Sijilli, by having the possibility to transport a health record in a protected USB stick, is considered a mobile EHR system that can be implemented in conflict-affected areas that do not have a digital infrastructure [31]. HealthEmove and My Personal Health Bank are patient-centric health record systems and primarily focused on providing patients with access to their health records. HealthEmove also contains a ‘library’ with information regarding care access [16,36,41].

##### Partnerships, Ownership, and Funding

Almost all initiatives are owned by, or operated by, not-for-profit organizations that rely on financial support from donors. The e-MCH Handbook was developed by UNRWA in collaboration with the Japan International Cooperation Agency (JICA) [30,35]. The HERA App, the only available open-source tool, was developed by HERA Digital Health and incubated at the Harvard Innovation Labs, in partnership with the Turkish Ministry of Health, United Nations agencies, and international humanitarian donors. It received funds from Grand Challenges Canada, the European Investment Bank, and Google [37]. Sijilli, results from a collaboration between the American University of Beirut and EPIC Health Systems, a for-profit US-based healthcare software company [28,31]. HealthEmove was initiated by Amsterdam Health & Technology Institute (Ahti), a not-for-profit institute, and utilizes the software of Patients Know Best (PKB), a UK-based for-profit company. PKB additionally functions as a Dutch PGO (Personal Health Environment) [36]. My Personal Health Bank, a collaboration between the University of Southern Denmark, the University of Dodoma, and the Muhimbili University of Health and Allied Sciences in Tanzania, received funding from Innovation Fund Denmark, Microsoft, a private investor, and INNOWWIDE [41]. RedSafe is part of the International Committee of the Red Cross (ICRC) and is funded by voluntary contributions from donors [35,38]. No information is available about specific funders of HealthEmove.

#### 3.2.2. Medical Information and Data Management

##### Health Data Stored in the Tools

Sijilli, My Personal Health Bank, HealthEmove, and the e-MCH Handbook are comprehensive health records, including medical history, diagnosis, medicine lists, and clinical measurements (e.g., blood pressure) [16,41]. My Personal Health Bank and Sijilli, specifically, include vaccination history [31,41]. In the e-MCH Handbook, the medical record is stored for both mother and children [27,30]. HealthEmove additionally includes clinical results, symptoms, imaging, and an audio function to record consultations or personal messages that can be stored in the health record [36]. In the HERA App and RedSafe, the users add medical documents and information to the “digital vault” and in this way decide which information they allow to be stored in the tool [37,38]. The apps thus serve as a repository for medical documents. In the HERA App, users can also keep track of vaccinations of their children and antenatal checkups [32].

##### Data Entry

The e-MCH Handbook is linked to UNRWA’s EHR system, filled out exclusively by HCPs [30]. No information is available about the patient’s ability to add medical information to the e-MCH Handbook. In Sijilli, data entry is restricted to authorized data entry personnel who collect data using data entry software run on ‘tablet computers’. External HCPs can add clinical notes online at the time of their encounter with the patient, using the cloud-based version of the Sijilli EHR [31]. The Sijilli record does not allow patients or HCPs outside of their system to edit any section of a patient’s health record [34].

Exclusively patients can add health data to the HERA App and RedSafe, in the form of documents and images of medical information uploaded to the ‘digital vault’ [37,38].

For HealthEmove data entry in the health record is allowed for both HCPs and patients. HCPs can add medical information to the patient record via the dedicated HCP portal. Both patients and HCPs can add pictures and documents to the record. For each data entry point, it is visible whether it was added by the patient or a specific HCP. Moreover, all data entered into HealthEmove is labelled into one of four categories: ‘general health’, ‘social care’, ‘mental health’, and ‘sexual health’ [36]. For My Personal Health Bank, the patient invites an HCP with a profile linked to a healthcare facility to insert health information into their record. However, in contrast to HealthEmove, patients using My Personal Health Bank can only enter information on relations (family members), date of birth and their name in the app but not medical information [unpublished data].

##### Data Sharing

For the e-MCH Handbook, data are automatically transferred between the UNRWA’s EHR system and the e-MCH app [30].

For My Personal Health Bank and HealthEmove, the patient can grant HCPs access to the health record. Patients using HealthEmove can invite a HCP via email to create an account and access their data, also other people such as family members can be invited via email. Moreover, they can decide which data they want to share with HCPs, based on the four labelling categories [36]. For My Personal Health Bank, HCPs in a facility can send a ‘connection request’ to the patients, who grant HCPs access to their information [unpublished data].

For RedSafe, within the ‘digital vault’, there is a sharing button (website Red Cross); however, information on how to share medical documents is not available. This information is also not available for the HERA App.

Sijilli was developed to give different levels of access to different users of the tool, including HCPs and patients. The USB stick that patients receive after data collection can be transported throughout their migration and used in any health facility around the world without restriction. Additionally, the cloud-based version of a Sijilli health record can be accessed globally, either by the patient or a healthcare provider, through the Sijilli website [28,34].

##### Data Storage, Security, and Offline Accessibility

All tools make use of a cloud unit to store medical information, except for the HERA App. My Personal Health Bank uses Azure cloud storage [41], and HealthEmove uses Google Cloud via their software provider Patients Know Best, having a local server in Amsterdam [36]. RedSafe uses its own ICRC servers to store medical information [38]. No information was available on the specific cloud storage of Sijilli.

Information on additional security measures remains limited in the literature. Claims on data security for the HERA App are restricted to using encrypted data and storing it in a decentralized location [37]. My Personal Health Bank uses a secure login procedure for accessing data on multiple devices and follows Microsoft data protection commitments [41]. RedSafe is governed by ICRC data protection rules, securing all personal data provided to the ICRC for humanitarian purposes. All data are encrypted and unreadable for external parties. Additionally, if users delete their accounts, the content of the ‘digital vault’ is also deleted [38].

For Sijilli, after data collection, an encrypted de-identified version of the generated health record is uploaded to the Sijilli server [28,31]. This cloud-based version of the health record can be accessed via the Sijilli website following a two-step identification process server [28,31]. All data from the data entry software can be wiped out after being synchronized [31]. The health record stored on the USB stick is protected by a password made of a unique combination of the patient’s personal information [28,34].

HealthEmove states on its website that it complies with the ISO 27001:2022 accreditation for managing information security and with the European Data Protection Act [36]. Users can request to delete their account via HealthEmove, who in turn forward the request to Patients Know Best [36].

The user applications of Sijilli, e-MCH Handbook, and RedSafe can operate offline. For the Sijilli tool, data can be added by HCPs without internet connectivity [31]. Also, mothers and pregnant women using the e-MCH Handbook can use the app offline, and data are immediately updated when the device is connected to the internet [27]. For RedSafe, documents can be downloaded from the ‘digital vault’ to a device that will be accessible offline [38]. Since both HealthEmove and My Personal Health Bank are web apps, internet access is required to retrieve and share medical information. This applies both to connection with the HCP portal and access to the web app itself. However, users of HealthEmove can download images and documents on their devices for offline availability [36].

#### 3.2.3. User Experiences of the Tools Identified (Including Health-Related Outcomes)

User experiences of the tools identified were only reported for the e-MCH Handbook, the HERA App, and My Personal Health Bank.

For the e-MCH Handbook, reported information on user experiences was first retrieved in 2017 [30,40]. Outcomes from a 2017 focus group with 22 pregnant women and mothers (with children aged 0–5), reported that, although most participants owned smartphones, they preferred the paper-based MCH Handbook, due to a lack of familiarity and navigation difficulties in the digital version [40].

Nasir et al. (2020) investigated factors associated with the dissemination and implementation of the e-MCH Handbook. Among the 1042 pregnant women and mothers with children aged 0–5 included in the analysis, 51.3% knew about the e-MCH Handbook, 23.8% downloaded it, and 17.4% used it. Among participants who were aware of the App, the presence of other apps on their mobile phones, staff knowledge of the e-MCH Handbook, and the use of the internet as a source of medical information were associated with downloading the e-MCH Handbook. Findings were limited to Android users due to technical issues with the iPhone version. Nasir et al. reported that the dissemination strategy (posters and pamphlets) and implementation strategy (staff training) were insufficient to promote user uptake of the e-MCH Handbook. Moreover, it was described that only 927 pregnant women and mothers downloaded and activated the e-MCH Handbook from the 200.000 eligible individuals. According to the authors, this highlights the scepticism of users towards mHealth technology [30].

Additional evidence on user experiences of the e-MCH Handbook was collected between 2019 and 2022 [35]. Based on preference data, this study reported that users primarily sought treatment, appointments, and health advice when using the e-MCH Handbook [35].

For the HERA App, the latest user experiences were captured in 2019. A qualitative study involving semi-structured interviews with 14 Syrian refugees (aged 19–37) who were pregnant or had children under two years reported generally positive feedback about the ease of use of the HERA App. The photo-taking feature of storing health records digitally was initially unclear, but interest increased after its purpose was explained. Participants valued the vaccination reminders and health information features, with most finding the tool easy to use. They appreciated the availability of information in Arabic and the location finder for nearby healthcare facilities, which was the most favoured feature. Suggestions for improvement included expanding health information, particularly on postpartum depression and mental health during perinatal care. Concerns about privacy and data protection were raised, particularly regarding data retrievability if the device was replaced. Additionally, some users faced issues with limited phone memory [33]. Moreover, a feasibility study was conducted in 2018 among 200 Syrian refugee pregnant women or mothers with children under the age of two in Istanbul. It showed that automated reminders for antenatal visits and childhood immunizations improved compliance, were well received, and offered a cost-effective alternative to other methods. It also confirmed that these women had smartphones and used them to access health information. No information was found on the specific methodology of this feasibility study [29].

For My Personal Health Bank, a feasibility study was performed in Tanzania between June 2022 and February 2023 [41]. In total, 4086 patients were included in My Personal Health Bank, and 1182 patients answered follow-up questions. Six hospitals and health centres were included. No more information is available on the methodology used. In total, 51.9% of the patients indicated that they did not have access to a smartphone. A large majority of the patients agreed with the following statements: ‘I will recommend MPHB to other people’, ‘MPHB makes me involved in my treatment’, ‘I feel confident that my health data are secure in MPHB’. However, while the majority agreed with the statement ‘I am willing to pay for MPHB’, when asked about the amount they were willing to pay, most participants chose ‘no payment’. Finally, even though information on the number of people actively using MPHB was not provided, reasons for not using MPHB included ‘I don’t have a smartphone’, ‘I forget about it’, ‘I’m not used to using apps and smartphone’ and ‘I don’t have money to buy internet’. Moreover, this study concluded, primarily based on the experiences of clinicians, that MPHB improved care and course of treatment for patients and is more time-efficient for doctors and patients.

#### 3.2.4. User Engagement in Tool Development

Information about user engagement in the development process of the tools was explicitly mentioned for Sijilli, the HERA App, and HealthEmove. For Sijilli, the design was tailored to the needs of stakeholders and users, including the Sijilli web portals. Stakeholders included physicians, nurses, community health workers, medical students, and medical residents [31]. A weakness reported on Sijilli was that the tool had limited customization functionality and required trained developers to customize the platform [34]. HealthEmove emphasizes on its website that co-creation is a core value that guides the development of the tool in collaboration with the community, although specific details about community engagement are not provided [36]. In the case of the HERA App, an agile methodology framework was employed, incorporating an iterative improvement process that integrated user feedback on the apps’ features [29].

## 4. Discussion

This systematic review specifically focuses on currently existing Electronic Personal Health Records (EPHRs) developed for and used by mobile populations. In total, six EPHRs were identified, described in only 10 scientific publications and additional grey literature sources, highlighting the limited body of research within this field. Nevertheless, the potential of EPHRs for mobile populations is recognized in the literature [6,16,21,24,53] and is in line with the WHO’s aim to ensure that refugees and migrants benefit from universal health coverage, which includes supporting countries in building accessible and culturally sensitive health systems [54].

While this study started with a broad definition of ‘mobile populations’, in practice, the initiatives identified were solely aimed at refugees, ‘people on the move’, migrants, and people affected by conflict, migration and other humanitarian crises.

### 4.1. Discussion of Key Findings

#### 4.1.1. Centralization of Health Records and Other Functionalities

The tools included in this review are at various stages of development, ranging from prototypes being piloted to tools in more mature phases of implementation. However, data available on facilitators and barriers to implementation is limited. The tools vary in how they centralize health records. Only Sijilli, My Personal Health Bank, HealthEmove, and the e-MCH Handbook are comprehensive health records, while the HERA App and RedSafe allow users to store medical pictures and documents in a ’digital vault’. Some of the tools include additional functionalities, such as access to health information, appointment reminders, chat functions, and educational material. The importance of adequate information about access to care for refugees is outlined by Chiarenza et al. (2019) who illustrate that for refugees, the lack of knowledge of entitlements and available health services is the most important obstacle to accessing care [7]. Moreover, the use of mobile phones in health services is commonly viewed as effective to, for example, schedule medicine reminders and medical appointments, among displaced populations [55]. In line with this, in the HERA App, vaccination reminders and health information features were valued most among users. Similarly, in the e-MCH Handbook users sought treatment, appointments, and health advice the most. This illustrates the potential of EPHRs to serve as more than just repositories of health records. It could also indicate that EPHRs may be more widely used if they integrate health-related reminders and/or features that facilitate access to care such as health information.

#### 4.1.2. Different Degrees of Patient Autonomy

Studies acknowledge that patient autonomy over health information is an important aspect of empowering individuals to actively engage in their healthcare management [6,21,53]. However, the degree of autonomy patients have in managing their health records varies across the tools examined in this review.

Patients have the most autonomy in managing their health data with RedSafe, the HERA App, HealthEmove, and My Personal Health Bank. In RedSafe, the HERA App, and HealthEmove patients can, for example, add health information to their own EPHR. Additionally, in HealthEmove and My Personal Health Bank patients must actively grant HCPs access to their health records [36,41], with HealthEmove even allowing patients to only give access to a subset of their health information [36]. Sijilli and the e-MCH Handbook give patients a more passive role in maintaining their own health records, as the EPHRs are connected to the EHR systems of HCPs, and information is added to their EPHR automatically. The importance of patients managing their health data was highlighted in a study by Damen et al. (2022) where it was found that patients who managed their own medical information felt more informed, experienced a greater sense of ownership over their healthcare, and demonstrated improved treatment adherence, compared to those who did not engage in managing their own health information [22]. Moreover, in a study by Tensen et al. (2025), HCPs argued that an EPHR could empower UDMs with more autonomy over their healthcare [53].

On the other hand, individuals do not always wish to be responsible for their own health record keeping, and having responsibility for their own health data can also impose a burden on individuals [22,53]. This burden might be even more significant for certain groups of mobile populations, which due to their migratory context must prioritize other needs to the detriment of their healthcare [53]. Moreover, it is important to consider that a higher level of autonomy requires a certain level of digital (health) skills from an individual. For example, with HealthEmove and My Personal Health Bank, the user must perform digital tasks to invite a healthcare provider to access their record.

Overall, while an EPHR has the potential to improve data exchange along migration routes—given that the patient remains the only ‘constant’ factor in this process—it is essential to gain a deeper understanding of the specific needs of mobile populations in relation to the level of autonomy they want over their health records and the necessary digital skills that are required to effectively navigate and use EPHRs.

#### 4.1.3. User-Adoption Remains a Critical Challenge

Even though limited data are available on user adoption of EPHRs, the evidence gathered in this review indicates that user adoption remains a critical challenge in the implementation of EPHRs. The study by Nasir (2020) on the e-MCH Handbook revealed that out of over 200,000 eligible individuals, only 927 pregnant women and mothers downloaded and activated the application in 2017. This low uptake was partly attributed to technological scepticism among users [30].

A key factor influencing user adoption is the engagement of users during the development process [24,56]. Gonzalez et al. (2021) demonstrated that participatory design and co-creation of mHealth tools foster long-term engagement [57]. Similarly, Jang et al. (2018) highlighted that user participation enhances access and enrolment in digital interventions [58]. Despite this evidence, the present review identified only three EPHR tools explicitly mentioning stakeholder involvement during development. Therefore, it is essential to invest in the active engagement of users and representatives from target groups, across all stages of EPHR design, implementation, and evaluation [24].

To make use of the tools identified in this research, mobile populations need to have access to digital means like internet connectivity and mobile devices. The HERA App, the e-MCH Handbook, and RedSafe require a smartphone to access health records. The other tools are available as web apps which people can access from a device connected to the internet. While mobile and internet connectivity are considered essential for mobile populations, not all have access to the internet when in need of healthcare [24]. Moreover, a lack of charging stations and or locations where internet infrastructure is unreliable are common in the living contexts of mobile populations [24]. The HERA App, RedSafe, and the e-MCH Handbook incorporate offline functionality, allowing users to access their tools without an internet connection. This feature is highly recommended, as it ensures that mobile populations can still engage with their health records even in areas with limited or no internet access.

For satisfactory user adoption of EPHRs, users also need to have a certain level of digital literacy, the level depending on the specific tools. Digital health literacy is often considered a challenge for migrants and refugees [59,60,61]. In this review, only the HERA App, the e-MCH Handbook, and My Personal Health Bank reported on user experiences, including digital health skills. Among 14 participants using the HERA App, most found the tool easy to use, but some said that they did not understand it and would need more time to learn how to use it [33]. User experiences on the e-MCH Handbook illustrated that while most participants owned a smartphone, they preferred the paper-based MCH Handbook, citing unfamiliarity and navigation difficulties with the e-MCH Handbook [40]. A systematic review on digital health among culturally and linguistically diverse populations reported that fundamental digital skills required to open an application and manually enter data are challenging for many individuals. This study also conveyed that fearing that technology is intimidating or dangerous was commonly reported [61]. This highlights the importance of educating users on the usage of EPHR tools [61] including on the content of EPHRs. Moreover, given the wide variety of digital health skills of potential end-users, an EPHR should not be a prerequisite for care, but rather a supportive element in care provision [53].

#### 4.1.4. Data Security Is a Key Priority for EPHRs Creators and Users

The security of the health data in an EPHR should be a key priority in its development [6,21,24,53], not only to comply with data protection legislation such as the EU General Data Protection Regulation, but also to ensure that health data, sometimes of sensitive character, is kept confidential and cannot be misused [21]. Data security is particularly important for groups of mobile populations, such as refugees, asylum seekers, and UDMs as they might find themselves in a political and legal context that makes them more vulnerable to surveillance and privacy breaches [62]. Mass media news, for example, show that authorities may unlawfully confiscate mobile phones to check identities and travel history [63,64,65].

While all EPHRs emphasize high data security in their design, this review also underlines that data safety can be a significant concern for users and can negatively influence user adoption of EPHRs. This was expressed by users of the HERA App who reported concerns about data security and whether the application shared their personal data with others [33]. The option of selecting which health data to share with HCPs, as featured in HealthEmove, might be an important way to address patients’ scepticism about storing sensitive health data in an EPHR and sharing it with HCPs.

All tools identified in this review use a cloud to store health data, except the HERA App. Cloud-based storage can mitigate risks associated with storage in physical devices such as loss or damage, which jeopardize health data availability. This is particularly important for mobile populations, who due to their mobility, might lose access to their mobile devices [66]. Nevertheless, centralized cloud-based storage is not inherently a secure way to store data, given the risk of accessing medical records due to breaches or misuse by authorities. It is important to explore secure, and decentralized networks, to best protect the sensitive data of mobile populations, and discover how mobile populations themselves can become owners of their health data [24]. Moreover, it is important to explore the compliance of technology, such as blockchain based technology, with EU data protection laws [67].

### 4.2. Implications for Research and Practice

This review highlights several critical gaps and opportunities for advancing research and practice related to EPHRs for mobile populations.

This study focussed on mobile populations, which encompass a wide variety of mobile groups. However, in practice, the identified EPHRs were focused on refugees, migrants and people affected by conflict, migration and other humanitarian crises. It is interesting to explore how EPHRs might benefit other subgroups of mobile populations, and how insights can be transferable across distinct groups.

The positive impact of EHRs on chronic disease management for migrants has been documented in the literature [25] and—in theory—patient-centric EPHR initiatives could improve health outcomes for mobile populations [6,16,21,53]. However, this review found no evidence to support this. While HCPs in a feasibility study on My Personal Health Bank indicated that using this EPHR is time-efficient for both patients and HCPs, none of the studies included in this review evaluated the impact of EPHRs on health outcomes nor on the continuity and quality of healthcare for mobile populations. This represents a significant area for further investigation.

Furthermore, data on user experiences has only been reported for three of the tools included in this review, underscoring the need to explore the perspectives of mobile populations and HCPs as end-users of EPHRs. Given the diversity of mobile populations, the variations within these groups, and the distinct contexts in which they live, such exploration is critical to developing user-centred and context-sensitive solutions. This underscores the likelihood that a single EPHR may not adequately address the needs of all mobile populations or even all individuals within a specific group.

Moreover, while almost all tools intend to accommodate cross-border data sharing, this is a highly under-researched topic. Currently, only two EPHR tools in this study operate across multiple countries. Besides potential language barriers when patients share their records across borders, a specific concern relates to data protection of sensitive health data, particularly because the social and legal context influencing disclosure of sensitive health data—such as contraception, sexually transmitted infections, and abortion—varies globally [21]. It is crucial to explore user experiences regarding cross-border sharing and discover what users think about specific mechanisms for health data protection, such as the potential for individuals to control access to specific parts of their health records. Tools like HealthEmove offer this functionality. However, further research is needed to assess the acceptability and usability of these features in practice.

Finally, a key challenge for multiple EPHRs for mobile populations is their sustainability, due to the dependency on donor funding, given that most tools are not-for-profit. It is important to conduct cost-effectiveness studies, to further explore the affordability of both developing and using EPHRs. The broader societal and health system benefits of EPHRs should also be demonstrated to attract funding or investments from governmental, philanthropic, and social impact funds and improve the scalability and availability of EPHRs.

Addressing these gaps will require a multidisciplinary approach, integrating perspectives from mobile populations, HCPs, policymakers, and technologists to ensure that EPHRs are both effective and equitable in meeting the unique needs of diverse mobile populations.

### 4.3. Strengths and Limitations

This rapid systematic review has several strengths and limitations. The review benefited from an interdisciplinary and international approach, with contributors from multiple countries and academic faculties. This diversity strengthened the study by incorporating a wide range of methodological perspectives and subject-matter expertise. Furthermore, two key experts in the field were consulted to validate the findings and ensure that no relevant initiatives were overlooked. Other strengths included that no restrictions were set on study design and geographic location to capture all relevant existing tools. Furthermore, grey literature was added to this review, to ensure an updated overview of current existing EPHRs for mobile populations. This has specifically allowed us to report on the existence of two EPHRs, which could not yet be found in peer-reviewed literature at the time of our search. Nevertheless, this review also has some limitations. Moreover, a limitation of this study is the exclusion of papers that are not written in English. This could have led to missing relevant papers written in other languages. Scarce data on implementation strategies and user experiences limited our ability to report on barriers and facilitators of implementation, and on the involvement of mobile populations in the design of EPHRs intended to address their needs. Due to the small number of relevant publications, the even sparser number of scientific studies, and the limited information on methodology in some of them, we did not exclude studies based on a quality assessment or risk of bias. Moreover, on two tools, only grey literature was included, which lacks scientific evidence. Therefore, it is important to acknowledge that we could not entirely assess the robustness of the evidence underlying all the literature included in this review.

## 5. Conclusions

In the current absence of adequate medical exchange systems across countries and across local healthcare facilities for mobile populations, individuals themselves can be considered the consistent entity in ensuring the continuum of care. The use of EPHRs to improve medical data exchange and thereby improve the continuity of care for mobile populations is promising; however, the field of research on EPHRs for mobile populations is still underexplored, with limited scientific data on their development, implementation, and impact. This rapid systematic review identified six EPHR tools designed for and used by mobile populations. The tools vary in how they centralize health records, ranging from ‘digital vaults’ to comprehensive systems, from smartphone apps to web-based apps, and from offline functionalities to the necessity of being connected to the internet. Moreover, they vary in the degree of autonomy they offer patients.

Key elements for further development and implementation of EPHRs include ensuring a high level of data security to ensure data protection of sensitive health data, educating users both on the usage and content of their EPHR and designing an easy-to-use and intuitive system. Moreover, EPHRs intended for cross-border use, should ensure user autonomy over which information is shared and explore ways to appropriately deal with sharing culturally sensitive information. Actively involving mobile populations in the development, implementation, and evaluation process is a crucial step toward enhancing the usability and adoption of EPHRs. Finally, it is important to recognize the diversity within mobile populations and the varying contexts in which they live, when developing user-centred, context-sensitive solutions to ensure that EPHRs effectively meet the diverse needs of mobile populations.

## Figures and Tables

**Figure 1 ijerph-22-00488-f001:**
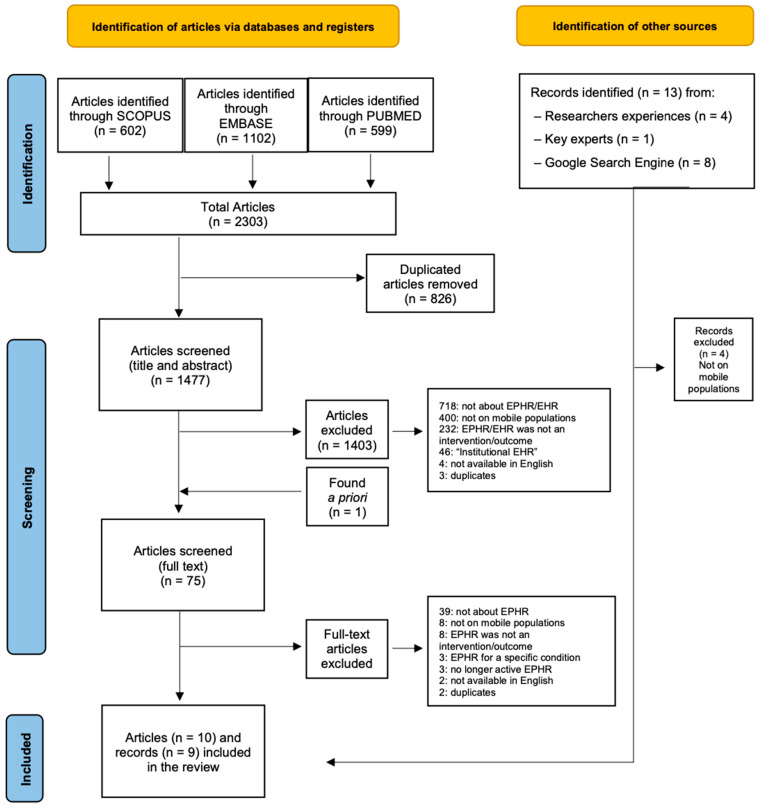
PRISMA 2020 flowchart for the inclusion of articles.

**Table 1 ijerph-22-00488-t001:** Main characteristics of the 10 eligible articles included in this review.

No.	Authorship (Year)	Title	Publication	Methodology	EPHR
1	Ballout, G. et al. (2018) [27]	UNRWA’s innovative e-Health for 5 million Palestine refugees in the Near East	Original article	NA	e-MCH Handbook
2	Saleh, S. et al. (2019) [28]	Sijilli: a mobile electronic health records system for refugees in low-resource settings	Comment	NA	Sijilli
3	Narla, N. et al. (2020) [29]	Agile application of digital health interventions during the COVID-19 refugee response	Viewpoint	Exploratory evaluation	HERA App
4	Nasir, S. et al. (2020) [30]	Dissemination and implementation of the e-MCH Handbook, UNRWA’s newly released maternal and child health mobile application: a cross-sectional study	Original article	Cross-sectional study design	e-MCH Handbook
5	Saleh, S. et al. (2020) [31]	Sijilli: A scalable model of cloud-based electronic health records for migrating populations in low-resource settings	Viewpoint	NA	Sijilli
6	Surmeli, A. et al. (2020) [32]	Leveraging mobile applications in humanitarian crisis to improve health: A case of Syrian women and children refugees in Turkey	Report	NA	HERA App
7	Meyer, C. et al. (2022) [33]	Perceptions on a mobile health intervention to improve maternal child health for Syrian refugees in Turkey: Opportunities and challenges for end-user acceptability	Original article	Qualitative study	HERA App
8	Shrestha, A. et al. (2022) [34]	Innovation is needed in creating electronic health records for humanitarian crises and displaced populations	Opinion paper	NA	Sijilli
9	Vijver, S. et al. (2023) [16]	Digital health for all: How digital health could reduce inequality and increase universal health coverage	Viewpoint	NA	HealthEmove
10	Seita, A. et al. (2024) [35]	Leveraging digital health data to transform the United Nations Systems for Palestine refugees for the post pandemic time	Original article	Qualitative study	e-MCH Handbook

**Table 2 ijerph-22-00488-t002:** Information on the grey literature sources included in this review.

No.	Authorship (Month, Year)	Title	Publication Type	EPHR Tool
1	HealthEmove (n.d.) [36]	Your Personal Health Record	Official website HealthEmove	HealthEmove
2	Hera Digital Health (n.d.) [37]	HERA Digital Health	Official website HERA Digital Health	HERA App
3	ICRC (n.d.) [38]	RedSafe, a Digital Humanitarian Platform	Official website ICRC	RedSafe
4	ICRC (February, 2022) [39]	Fourth RedSafe kiosk opens in Zimbabwe	News release on the official website ICRC	RedSafe
5	JICA (August, 2017) [40]	Jordan: UNRWA’s electronic MCH handbook application for Palestine refugees (Issue 20)	Technical brief	e-MCH Handbook
6	My Personal Health Bank (n.d.) [41]	My Personal Health Bank	Official website of My Personal Health Bank	My Personal Health Bank
7	My Personal Health Bank (n.d.) [42]	Usage statistics	LinkedIn post	My Personal Health Bank
8	MIT-solve (August, 2022) [43]	Novel measurement for performance improvement challenge My Personal Health Bank	Application MIT-solve	My Personal Health Bank
9	Play Store (May, 2024) [44]	Play Store: RedSafe	App download	RedSafe

ICRC, International Committee of the Red Cross; JICA, Japan International Cooperation Agency; n.d., no date.

**Table 3 ijerph-22-00488-t003:** Description of the six EPHRs tools identified.

Tool Name	Initiated/Owned by	Mobile Population	Current Countries	Stage of Development	Number of Users (Month, Year) ^a^	Languages	Tool Description	Application Type
Tools from the scientific literature								
e-MCH Handbook	UNRWA and JICA ^#^	Palestine refugees ^#^	Jordan, Gaza, Lebanon, Westbank, Syria ^#^	Application since 2017 ^#^	254,586 registered users (July 2023) and 22,000 active users (June 2023) ^#^	Arabic ^#^	mHealth application with PHR	Smartphone app ^#^
HERA App	HERA Digital Health ^&^	Syrian refugees ^#^	Turkey ^#^	Application since 2018. In 2020 field tests were performed ^#^	>3000 refugee families in Turkish pilot study (n.d.) ^&^	Arabic, Turkish, English ^#^, Pashto and Dari ^&^	Humanitarian platform with ‘digital vault’	Smartphone and web-app ^#^
Sijilli	American University of Beirut and Epic Health Systems ^#^	Syrian refugees ^#^	Lebanon ^#^	Launched 2018 ^#^	>10,000 users (2022) ^#^	English and Arabic ^#^	EHR with user-portal and USB-stick	NA
HealthEmove	Initiative: Amsterdam Health & Technology Institute; software: Patients Know Best 5.0.18 ^&^	Refugees ^#^ and People on the move ^&^	Netherlands ^&^	Application since 2023 ^#^	NA	22 languages ^&^	EPHR	Web-app ^&^
Tools from the grey literature								
My personal Health Bank	University of Southern Denmark, University of Dodoma and Muhimbili University ^&^	People in developing countries and people on the move ^&^	Tanzania ^&^	Feasibility study from June 2022 until February 2023	4969 patients included (June 2022) ^&^ (31) ^a^	English, Kiswahili ^&^	EPHR	Web-app ^&^
RedSafe	International Committee of the Red Cross ^&^	People affected by conflict, migration and other humanitarian crisis ^&^	Honduras, Guatemala, El Salvador, Mexico, USA, Costa Rica, Panama, Zimbabwe, South Africa, Botswana, Malawi, Mozambique, Eswatini, Lesotho, Switzerland, Zambia *^,&^	Launched in May 2021 ^&^	32,000 downloads (February 2022) ^&^	English, Spanish and Portuguese ^&^	Humanitarian platform with ‘digital vault’	Smartphone and web-app ^&^

Tables may have a footer: JICA, Japan International Cooperation Agency * available for download ^a^ or number of downloads; ^&^ Information retrieved from grey literature; ^#^ Information retrieved from scientific articles.

## Data Availability

No new data were created in this study. Data sharing is not applicable to this article as it is based on a systematic review of publicly available literature.

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
