# Peer review of "Electronic Personal Health Records for Mobile Populations: A Rapid Systematic Literature Review"

_ijerph, 2025, doi:10.3390/ijerph22040488_

Round 1
Reviewer 1 Report
Comments and Suggestions for Authors
Given the vulnerability of healthcare in mobile populations globally, particularly in developing or less developed countries, the authors have addressed an important and much-needed new topic. Overall, the manuscript is of relatively good quality and logically sequenced. The following suggestions are offered to improve the quality of the manuscript.
- The time of study implementation should be stated in the abstract.
- Although the Embase database covers the articles of the Medline database, these databases are different and separate from each other. Therefore, using the two interchangeably is incorrect. Thus, the authors need to correct the inconsistency of the names of the databases in the abstract and methods section.
- Given the fundamental difference between EHR and EPHR in terms of data scope, data source, data ownership, management, etc., it is completely incorrect to use these two concepts interchangeably. The authors used EHR as the inclusion criteria for the inclusion criteria, while their study focused on EPHR. Why?
- The type of articles (original, review, etc.) is not specified in the inclusion and exclusion criteria section. The period of the year of publication of the articles is not specified.
- Given the type of study (rapid systematic literature review), and based on my knowledge of these studies, a quality assessment of the included articles should have been performed. Why was it not done in this study?
- It is recommended that the exact reason for excluding articles at each stage be briefly stated in the PRISMA flowchart.
- Appendices are usually placed after the references. It is better to consider this in this study as well.
Author Response
Comment 1: The time of study implementation should be stated in the abstract.
Response 1: Thank you for pointing this out. We have addressed this including the time period of this study in the abstract page 1 lines 32–34: “A rapid systematic review was conducted, between September 2024 and January 2025, identifying relevant publications through searches in Embase, PubMed, Scopus, and grey literature.”
Comment 2: Although the Embase database covers the articles of the Medline database, these databases are different and separate from each other. Therefore, using the two interchangeably is incorrect. Thus, the authors need to correct the inconsistency of the names of the databases in the abstract and methods section.
Response 2: Thank you for this comment. We agree that it is not correct to use Embase and Medline interchangeably. We have replaced the term “Medline” with “Embase” on page 4 lines 159–161: “The query was run on three scientific databases (EMBASE, PUBMED and SCOPUS), in October 2024, for articles published between 2014 and 2024.”
Comment 3: Given the fundamental difference between EHR and EPHR in terms of data scope, data source, data ownership, management, etc., it is completely incorrect to use these two concepts interchangeably. The authors used EHR as the inclusion criteria for the inclusion criteria, while their study focused on EPHR. Why?
Response 3:
Thank you for this important remark. We agree that it is not correct to use EHR and EPHR interchangeably and we did not intend to do so. We have previously phrase our criteria based on EHRs because we wanted to be sure that we would include any kind of electronic health record with a personal component to data management. As the field of EPHRs is still evolving we assumed that there could be tools with a personal component to data management, though not identified as EPHRs per se in publications’ titles and/or abstracts.
However, as pointed out by the reviewer, our study focuses on EPHRs for mobile populations and after full-text screening we retained only articles describing EPHRs, with a digital tool that allows patients to manage and access their health records. We have clarified our process by removing the reference to EHR in the inclusion/exclusion criteria, and explaining that our screening became more narrow during full-text screening where often more detailed information on the personal component of tools was provided. Our revised description of the inclusion/exclusion criteria can be found on page 4 lines 176 – 190: "The inclusion and exclusion criteria for articles in this review were defined a priori. Articles were included if they (1) were published in English; (2) were published between January 2014 and October 2024; (3) described an EPHR for mobile populations; (4) explored EPHRs as an intervention or outcome; (5) described EPHRs which were still existent; (6) described EPHRs which were not restricted to a specific medical condition; and (7) described EPHRs which included a digital component that allowed the patient to manage their health records. This set of criteria led to the exclusion of publications if the described tool was only used as a data source to collect quantitative outcomes for a study, if the tool was an existing facility-based system and if the tool was developed mainly for HCPs in a specific health clinic. Moreover, studies were excluded when no full text was available. Inclusion criteria were applied sequentially. To include any tool having a personal component to health data management, but perhaps not named as an EPHR in the title/abstract, we retained articles referencing to EHR in their title/abstract, and narrowed our inclusion criteria during full-text screening once enough detail was provided to match our focus on EPHRs."
Comment 4: The type of articles (original, review, etc.) is not specified in the inclusion and exclusion criteria section. The period of the year of publication of the articles is not specified.
Response 4: Thank you for mentioning these two points.
The year of publication is stated as follows in page 4 lines 159 –161: “The query was run on three scientific databases (EMBASE, PUBMED and SCOPUS), in October 2024, for articles published between 2014 and 2024”. However, we agree with your comment that this time period should be mentioned as an inclusion criteria. For clarity, we have also included the information in page 4 lines 177–178: “Articles were included if (1) they were published in English; (2) they were published between January 2014 and October 2024;”.
We also agree that the types of articles we included or excluded are missing. To address this we have added the following information in page 4 lines 190–192: “Literature reviews were excluded, but no further restriction on the type of article was applied. Articles included in literature reviews that met our inclusion criteria were screened individually, to make sure that relevant articles were not missed.”
Comment 5: Given the type of study (rapid systematic literature review), and based on my knowledge of these studies, a quality assessment of the included articles should have been performed. Why was it not done in this study?
Response 5: Thank you for this important comment. We have thoroughly discussed with our research team the relevance of conducting a quality assessment of the 10 publications included in our rapid review. While we acknowledge the importance of assessing the bias of studies included in literature reviews, we decided to not conduct a quality assessment of the publications included in our review for the following reasons. Given that EPHRs are a rapidly evolving and emerging field, we sought to capture a broad range of literature, including both peer-reviewed articles and grey literature, to provide a comprehensive overview of the current landscape of EPHRs for mobile populations. Though a quality assessment of data from grey literature was not possible, we restricted our grey literature sources to “official websites of EPHR tools, government and university webpages, reports, technical notes and information derived from the Play Store on the selected tools” to guarantee an adequate level of data accuracy. Among the 10 peer-reviewed publications included in our rapid review, few of them correspond to scientific studies with a clear study design and methodology. A quality assessment of all included publications based on objective guidelines was therefore also not possible. Based on this diversity of data sources and types, we decided that a quality assessment of the scientific publications would not add significant value to our publication, as our aim is to provide a comprehensive overview on the existent EPHRs for mobile populations. Nevertheless, we underline in the discussion section that this needs to be considered when interpreting and referencing our results “it is important to acknowledge that we cannot entirely assess the robustness of the evidence underlying all the literature included in this review” (page 20 lines 739 – 740).
Comment 6: It is recommended that the exact reason for excluding articles at each stage be briefly stated in the PRISMA flowchart.
Response 6: Thank you for suggesting this revision of the PRISMA flowchart. The flowchart on page 6 of the manuscript has been updated to include the requested information.
Comment 7: Appendices are usually placed after the references. It is better to consider this in this study as well.
Response 7: Thank you for this suggestion. As recommended by the reviewer, we have replaced the appendices and they can now be found after the references.
Reviewer 2 Report
Comments and Suggestions for Authors
First of all, we would like to congratulate you on this work, as we believe it to be relevant and well-executed research. We therefore commend the editors for its publication.
However, we believe that there are certain aspects that could perhaps be improved. Among them:
- In the search strategies, it is a little surprising that a widely known and recognised database such as WOS is not used and that there is no explanation as to why this database is not used and other similar ones are used.
- In the selection of studies, a greater clarification of the exclusion criteria is lacking, because while the acceptance criteria are more or less adequately explained, the same is not done with the exclusion criteria. Likewise, given the geographical scope of the study and the type of population to be studied (irregular or mobile immigrants), we believe that using the English language as a search and exclusion criterion may limit the potential of the research, both because of the articles published in other languages that are discarded, and because of the languages used by the people to whom the study is addressed (irregular or mobile immigrants).
- Finally, it seems a little surprising that given the high number of articles identified (2303), after the application of the filters this number is reduced to only 10 articles.
We think the article is very interesting and the suggestions we make are not to change anything in your research but to explain or clarify some of these aspects we allude to.
Author Response
Comment 1: In the search strategies, it is a little surprising that a widely known and recognised database such as WOS is not used and that there is no explanation as to why this database is not used and other similar ones are used.
Response 1: Thank you for your comment. Based on the rapid nature of our literature review, and informed by literature, we have decided to restrict the number of databases included for our literature review. We chose to include Embase, Pubmed and Scopus to broadly cover publications within biomedical, life and health sciences, which we consider the best focus to capture most relevant papers to explore EPHRs for mobile populations. To the best of our knowledge, Scopus and WOS have a similar multidisciplinary research focus, and we decided to include Scopus in our search because of its larger database.
Comment 2: In the selection of studies, a greater clarification of the exclusion criteria is lacking, because while the acceptance criteria are more or less adequately explained, the same is not done with the exclusion criteria.
Response 2: Thank you for mentioning this lack of clarity, which was also brought up by another reviewer. Based on both comments, we have reviewed the description of the inclusion/exclusion criteria which now reads as follows on page 4 lines 176 – 192: "The inclusion and exclusion criteria for articles in this review were defined a priori. Articles were included if they (1) were published in English; (2) were published between January 2014 and October 2024; (3) described an EPHR for mobile populations; (4) explored EPHRs as an intervention or outcome; (5) described EPHRs which were still existent; (6) described EPHRs which were not restricted to a specific medical condition; and (7) described EPHRs which included a digital component that allowed the patient to manage their health records. This set of criteria led to the exclusion of publications if the described tool was only used as a data source to collect quantitative outcomes for a study, if the tool was an existing facility-based system and if the tool was developed mainly for HCPs in a specific health clinic. Moreover, studies were excluded when no full text was available. Inclusion criteria were applied sequentially. To include any tool having a personal component to health data management, but perhaps not named as an EPHR in the title/abstract, we retained articles referencing to EHR in their title/abstract, and narrowed our inclusion criteria during full-text screening once enough detail was provided to match our focus on EPHRs. Literature reviews were excluded, but no further restriction on the type of article was applied. Articles included in literature reviews that met our inclusion criteria were screened individually, to make sure that relevant articles were not missed."
Additionally we have added reasons for exclusion to the PRISMA Flowchart on page 6 of the manuscript.
Comment 3: Likewise, given the geographical scope of the study and the type of population to be studied (irregular or mobile immigrants), we believe that using the English language as a search and exclusion criterion may limit the potential of the research, both because of the articles published in other languages that are discarded, and because of the languages used by the people to whom the study is addressed (irregular or mobile immigrants).
Response 3: Thank you for this very important comment. We agree that this is a limitation in our study and that we could have missed relevant information by excluding papers not written in English. Therefore, we have added this limitation in our discussion on page 10 lines 752–754: “Moreover, a limitation of this study is the exclusion of papers that are not written in English. This could have led to missing relevant papers written in other languages”.
Comment 4: Finally, it seems a little surprising that given the high number of articles identified (2303), after the application of the filters this number is reduced to only 10 articles.
Response 4: Thank you for this comment. While we understand that this can be seen as surprising, the field of EPHR is a new field of research resulting in limited available studies that fit our inclusion criteria. Moreover, a significant number of publications was duplicated, and the number of articles identified is therefore 1477. Nevertheless this is still a large number, compared to the final 10 articles retained. As we clarified in response to a comment by another reviewer, our criteria for title/abstract screening targeted EHRs in general to be sure we would include any tool including a personal component to health data management, but perhaps not named as an EPHR in the title/abstract. This led us to identify a large number of articles. However, when screening the articles full-text, we narrowed our inclusion criteria to match our focus on EPHRs and only kept those publications that indeed described EPHRs. This led to us retaining only 10 publications. We have clarified this by revising the inclusion/exclusion criteria described on page 4 lines 176 – 190 of the manuscript: "The inclusion and exclusion criteria for articles in this review were defined a priori. Articles were included if they (1) were published in English; (2) were published between January 2014 and October 2024; (3) described an EPHR for mobile populations; (4) explored EPHRs as an intervention or outcome; (5) described EPHRs which were still existent; (6) described EPHRs which were not restricted to a specific medical condition; and (7) described EPHRs which included a digital component that allowed the patient to manage their health records. This set of criteria led to the exclusion of publications if the described tool was only used as a data source to collect quantitative outcomes for a study, if the tool was an existing facility-based system and if the tool was developed mainly for HCPs in a specific health clinic. Moreover, studies were excluded when no full text was available. Inclusion criteria were applied sequentially. To include any tool having a personal component to health data management, but perhaps not named as an EPHR in the title/abstract, we retained articles referencing to EHR in their title/abstract, and narrowed our inclusion criteria during full-text screening once enough detail was provided to match our focus on EPHRs."